# Low-Temperature Plasma Diagnostics to Investigate the Process Window Shift in Plasma Etching of SiO_2_

**DOI:** 10.3390/s22166029

**Published:** 2022-08-12

**Authors:** Youngseok Lee, Sijun Kim, Jangjae Lee, Chulhee Cho, Inho Seong, Shinjae You

**Affiliations:** 1Department of Physics, Chungnam National University, Daejeon 34134, Korea; 2Samsung Electronics, Hwaseong-si 18448, Korea; 3Institute of Quantum Systems (IQS), Department of Physics, Chungnam National University, Daejeon 34134, Korea

**Keywords:** low-temperature plasma, plasma diagnostics, plasma etching, plasma process modeling

## Abstract

As low-temperature plasma plays an important role in semiconductor manufacturing, plasma diagnostics have been widely employed to understand changes in plasma according to external control parameters, which has led to the achievement of appropriate plasma conditions normally termed the process window. During plasma etching, shifts in the plasma conditions both within and outside the process window can be observed; in this work, we utilized various plasma diagnostic tools to investigate the causes of these shifts. Cutoff and emissive probes were used to measure the electron density and plasma potential as indicators of the ion density and energy, respectively, that represent the ion energy flux. Quadrupole mass spectrometry was also used to show real-time changes in plasma chemistry during the etching process, which were in good agreement with the etching trend monitored via in situ ellipsometry. The results show that an increase in the ion energy flux and a decrease in the fluorocarbon radical flux alongside an increase in the input power result in the breaking of the process window, findings that are supported by the reported SiO_2_ etch model. By extending the SiO_2_ etch model with rigorous diagnostic measurements (or numerous diagnostic methods), more intricate plasma processing conditions can be characterized, which will be beneficial in applications and industries where different input powers and gas flows can make notable differences to the results.

## 1. Introduction

Plasma is defined as a quasi-neutral gas of charged and neutral particles that exhibits collective behavior [1,2]. The characteristic features that make plasma distinct from other discharge phenomena are utilized in many industrial and research fields in terms of, for instance, controlling the dynamics of the component particles for individual applications [1]. With an enormous range of electron densities and temperatures, which are the most representative parameters, plasma has characteristic physical and chemical properties depending on the electron density and temperature regimes, resulting in a diverse categorization of plasma including material processing plasma and fusion plasma [2].

With the rapid growth of the semiconductor industry in the 20th century, material processing has grown into one of the biggest sub-fields of low-temperature plasma, designated as such by its electron temperature regime [3,4]. The essential processes in semiconductor manufacturing, such as etching [5,6,7,8,9], deposition [10,11,12], cleaning [13,14], etc., widely employ low-temperature plasma, allowing plasma to play a large role in the microelectronics industry [3,15,16,17]. Further development of electronic devices, however, requires more advanced plasma techniques to meet market demands, thereby increasing the processing complexity and difficulty. In this circumstance, plasma diagnostics can provide qualitative and quantitative information on plasma parameters for an understanding of the physical and chemical phenomena in the plasma processes, giving rise to the development of plasma technology [18,19,20,21,22,23].

Obtaining internal plasma parameters via plasma diagnostics can significantly help plasma processing engineers to establish the process window, which can be defined as the condition of the processing equipment or plasma itself that has to be met to realize the purpose of the process. For instance, SiO_2_ etching with fluorocarbon (FC) plasma requires that plasma ions be sufficiently strong; otherwise, the FC plasma would form thick FC films on the SiO_2_ instead of etching it [24,25]. Another example can be found in a special plasma process called atomic layer deposition (ALD), where a specific temperature window is necessary to realize atomic-scale deposition without defect-producing chemical reactions such as condensation or desorption at temperatures below or above the window, respectively [26]. Similarly, atomic layer etching (ALE), the counterpart of ALD, also has a characteristic process window with respect to the appropriate ion energy range that achieves atomic-scale removal without defect-producing physical reactions such as insufficient removal or sputtering at ion energies below or above the window, respectively [27,28].

There are numerous reports on the demonstration of plasma diagnostics via various methods to achieve the process window [21,23,29,30,31,32,33,34]. Compared to continuous plasma processes where a single plasma is maintained throughout the processing time, certain plasma processes such as ALE, where two or more kinds of plasma are alternated step by step, may especially benefit from plasma diagnostics. One previous report covers a comprehensive investigation into the discharge physics of ALE plasma, from several fundamental plasma parameters such as electron density and temperature to discharge instability and recovery periods during the ALE process [35].

In this work, a process window shift in SiO_2_ etching with FC plasma from a variety of input power is investigated via plasma diagnostics, the tools of which are carefully considered for their appropriateness to the polymeric conditions of FC plasma. Based on a previous report [36] that a steady state of etching conditions is determined by the balance between FC film deposition and SiO_2_–FC film removal rates, which are reflected by FC radical and ion energy fluxes, respectively, FC radical density is considered to be the parameter indicating the FC radical flux is this work. The SiO_2_ etch model is expressed as follows [36]:(1)$$\frac{dx_{total}}{dt}=\frac{d\left(x_{FC}+x_{SiO_{2}}\right)}{dt}=DR_{FC}-ER_{FC}-ER_{SiO_{2}},$$
where $x_{total},\;x_{FC}\;\mathrm{and}\;x_{SiO_{2}}$ stand for the thickness of the total (FC film + SiO_2_), FC film, and SiO_2_, while $DR_{FC},\;ER_{FC}\;\mathrm{and}\;ER_{SiO_{2}}$ stand for the deposition rate of FC films, etch rate of FC films, and etch rates of SiO_2_, respectively.

This parameter is diagnosed in real time through quadrupole mass spectrometry with a comparison to etching results also obtained in real time via in situ ellipsometry. Meanwhile, the ion energy flux is estimated from the electron density measured by a cutoff probe, and the plasma potential is found with an emissive probe. These multiple diagnostic results support the understanding of the process window shift occurring from power variation, providing a guideline for external parameter controls for improved processing results. Details of the SiO_2_ etching process, as well as the experimental setup and methods for plasma diagnostics, are described below, followed by a discussion on the processing results based on the plasma diagnostics.

## 2. Experiment

### 2.1. Plasma Etching

In this work, SiO_2_ (iNexus, Inc., Seongnam-si, Korea) etching is conducted not in a continuous manner but rather via ALE. Continuous SiO_2_ etching with FC plasma, which normally employs capacitively coupled plasma sources to achieve high-energy ion bombardment for high etch rates, benefits from the synergetic effect of the simultaneous reactions of reactive FC radicals and high-energy bombarding ions on the material surface. On the other hand, in ALE, the reactions of the radicals and ions are separated to obtain atomic-precision etch control [27,28,37]. A comparison between continuous etching and ALE is illustrated in Figure 1. In continuous SiO_2_ etching, both etch gases (e.g., a mixture of Ar and C_4_F_8_) and radio-frequency (RF) power are employed simultaneously to maintain the processing plasma throughout the etch process.

Alternatively, the ALE process is divided into two steps: Surface modification and removal, as labelled A and B in Figure 1, respectively. In the surface modification step (A), C_4_F_8_ is injected into a continuous Ar plasma and then dissociates into diverse FC radicals such as C_2_F_4_ and CF_2_ [38,39], allowing an FC film to grow until the C_4_F_8_ injection is cut off. The following removal step (B) begins with the C_4_F_8_ cutoff as the Ar ion bombardment physically sputters the deposited FC film. This sputtering continues until both the FC film and SiO_2_–FC mixed layer are totally removed and the underlying SiO_2_ is exposed. This can only be achieved with well-controlled ion energies that are higher than the sputtering threshold energies of the FC films and the mixed layer but lower than that of SiO_2_, a criterion typically called the ion energy window. Satisfying the ion energy window in the removal step of ALE leads to self-limiting etching, meaning that the etching spontaneously stops with the exposure of the new SiO_2_ surface (as illustrated in Figure 1), which is the most fundamental characteristic of the ALE process.

An inductively coupled plasma source is employed in the present work, and the plasma chamber has a cylindrical geometry with a diameter of 330 mm and a height of 250 mm. The substrate on which the SiO_2_ samples are processed is separated from the ceramic plate by 100 mm. An ellipsometer is mounted to the chamber that allows in situ monitoring of sample thicknesses during the entire etch process. Further details of this ALE chamber setup are described in our previous report [36].

The sequence of the etch process of the present work is illustrated in the lower panel of Figure 1. C_4_F_8_ is injected in a pulsed manner into continuous Ar plasma for surface modification, followed by the C_4_F_8_ cutoff that leads to a removal of the modified surface by the Ar ion bombardment. RF power with a frequency of 13.56 MHz is applied, and 44 sccm of Ar is injected into the chamber, resulting in a pressure of 1906 Pa. The flow rate of C_4_F_8_ is 2 sccm, which barely changes the chamber pressure.

Figure 2a,b plots the results of plasma etching based on the ALE recipe described above at an RF power of 100 and 300 W, respectively. The surface modification and removal steps are clearly separated in Figure 2a, where the thickness increases with the C_4_F_8_ injection and then decreases and saturates after the C_4_F_8_ cutoff. Since the FC film deposition during each modification step and the self-limiting etch trend during each removal step are well produced, the ALE condition of Figure 2a is considered to be in the ALE window.

On the other hand, increasing the RF power from 100 W to 300 W results in significantly different trends, as shown in Figure 2b. Here, C_4_F_8_ injection into the continuous Ar plasma actually leads to continuous SiO_2_ etching with FC plasma rather than the FC film deposition (surface modification) of ALE. This reflects that an increase in RF power shifts the processing condition out of the ALE window, and thus plasma diagnostics should follow to determine the causes of this process window shift.

### 2.2. Plasma Diagnostics

It is important to determine appropriate diagnostic parameters to rigorously investigate plasma processing. In the present work, the target parameters are chosen based on the SiO_2_ FC plasma etch model, explaining that a change in the material thickness is determined by the balance between its increasing and decreasing rates [36]; for an FC film, the increase rate corresponds to the FC film deposition rate and the decrease rate corresponds to the sum between its physical sputtering and chemical etch rates with SiO_2_. Whereas, for SiO_2_, its increase rate is assumed to be zero since there is no SiO_2_ source during the etching, and the decrease rate is set to be the same as the chemical etch rate of the FC film. The dominant parameters related to the deposition and removal rates of the FC film are the FC radical flux and ion energy flux, respectively, which can be described with the specific internal plasma parameters of FC radical density, electron density, and plasma potential. Below, diagnostic data acquisition and processing methods, as well as the geometry of the diagnostic tools, are described in detail.

#### 2.2.1. Electron Density Measurement

Since electron density is one of the most fundamental plasma parameters, there have been numerous studies on the development of electron density diagnostics. Langmuir proposed a historic plasma diagnostic tool, eponymously named the Langmuir probe, that provided not only fundamental parameters such as electron density and temperature but also electron energy distribution functions [40]. This probe has contributed enormously to a deeper understanding of plasma dynamics and characteristics, and to date, still plays a crucial role in plasma research areas including plasma physics [18,41]. However, Langmuir probe diagnostics significantly deteriorate in the harsh environments of plasma in material processing such as etching and deposition and the use of various processing gas mixtures; severe polymer deposition on the probe tip interrupts the probe operation, and the complexity of processing gas mixtures gives rise to considerable theoretical errors during data processing [2].

Microwave plasma diagnostics have emerged as an excellent alternative to the Langmuir probe, especially for processing plasma [42]. The most noticeable feature of microwave diagnostics is that they are barely perturbed by contamination from polymer deposition in processing plasma, as well as by RF noises to which most electrical diagnostic tools such as the Langmuir probe are vulnerable [22]. Among various types of microwave diagnostic tools developed over the years, the cutoff probe was chosen for the present work due to its simplicity in manufacturing and utilization [43,44,45].

The physics of the cutoff probe measurement is the cut-off phenomenon in plasma [2,46]. Plasma has a characteristic electron oscillation frequency, normally termed plasma frequency, which is expressed in terms of electron density as follows:(2)$$f_{\mathrm{plasma}}=\frac{1}{2\pi }\sqrt{\frac{n_{e}e^{2}}{\epsilon_{0}m_{e}}},$$
where *n_e_* is the electron density, *e* is the elemental charge, $\epsilon_{0}$ is the vacuum permittivity, and *m_e_* is the electron mass. The higher the electron density, the higher the plasma frequency, enabling the plasma to act similarly to a metal. When an electromagnetic (EM) wave meets plasma, if the wave frequency is higher than the plasma frequency, then the wave will pass through the plasma as if it is a dielectric medium, but if the wave frequency is lower than the plasma frequency, it will not pass through since the plasma, in this case, acts as a conductive medium. This shows that finding the cutoff frequency of plasma will provide its electron density.

The geometry of the cutoff probe used in this work is illustrated in Figure 3a. Except for the vacuum components, a cutoff probe typically consists of two coaxial cables and a signal generator. The two cables play the roles of radiating and detecting antennas. In this work, the antennas are made by stripping one end of two sub-miniature version A (SMA) cables that have no connector by approximately 10 mm and separating them by approximately 3 mm to allow the plasma to fill the space between the antennas. A vector network analyzer (S33601B, SALUKI TECHNOLOGY, Taipei, Taiwan) is used as a microwave signal generator with frequencies ranging from hundreds of kHz to 8.5 GHz.

With such a cutoff probe measurement system, one can obtain the S21 spectrum that shows the cutoff frequency, the typical form of which is plotted in Figure 3b. The sharp increase and decrease in S21 before the cutoff frequency have been reported to originate from the resonance between the sheath capacitances and plasma inductance. The electron density can then be calculated from the obtained S21 spectrum using Equation (2) [47].

#### 2.2.2. Plasma Potential Measurement

Plasma potential is also one of the most fundamental plasma parameters. Although the plasma potential can theoretically be measured with the Langmuir probe, its use is limited in processing plasma diagnostics due to probe tip contamination, as explained above. As an alternative, an emissive probe was chosen in the present work. The working principle of the emissive probe is similar to that of the Langmuir probe, but the critical difference between them is that the probe tip of the emissive probe emits thermionic electrons by Ohmic heating, as its name implies. Probe tip heating impedes polymer deposition, allowing the emissive probe to endure the processing plasma environment.

The physics of the emissive probe is briefly described as follows [2,48,49]. A W wire immersed in plasma without being electrically connected to its surroundings other than the plasma has a floating potential since the sheath between the W wire and plasma is effectively filled with positive charges that keep the flux of positive ions from the plasma to the W wire equal to that of electrons from the plasma to the wire. As illustrated in Figure 4a, a floating DC power supply connected to the W wire produces a conduction current that gives rise to Ohmic heating in the W wire, leading to thermionic electron emission. The positive and negative potential of the W wire is measured with digital multimeters (101, FLUKE, Washington, DC, USA) to estimate the probe potential at the center where the Ohmic heating is the strongest. Figure 4b shows an example of the emissive probe data of the change in the floating potential of the W wire as a function of the applied DC voltage, or heating voltage. When the heating voltage is significantly low, thermionic emission barely occurs, and the probe potential remains unchanged (see region (i) in Figure 4b). As the heating voltage increases, thermionic electrons start to affect the floating sheath potential, leading to an increase in the probe potential (regime (ii)). When the heating voltage is sufficiently high, the emission current becomes balanced with the plasma electron current and the probe potential levels off at the plasma potential (regime (iii)). In short, with a sufficiently high heating voltage, the plasma potential can be obtained by simply measuring the average of the positive and negative potential of the thermionic electron-emitting W wire.

#### 2.2.3. FC Radical Density Qualitative Measurement

FC radical densities such as CF_2_ and CF_3_ are measured with a quadrupole mass spectrometer (QMS) (PSM, Hiden Analytic, Warrington, PA, USA). Intensively developed over decades, QMSs have been widely used for gas-phase species diagnostics in plasma [50]. A QMS, also known as a residual gas analyzer, measures the partial pressures of each gas-phase species in a vacuum [51]. It is normally equipped in the main chamber via an orifice with a diameter on the micrometer scale, which allows the QMS chamber to maintain a higher vacuum level than that of the main chamber so that particles transiting in the QMS arrive at the detector with no collision.

The typical components of a QMS are an ionizer, a mass filter, and a detector [50]. In QMSs, gas-phase neutral species should be ionized before entering the mass filter, which is a quadrupole with two pairs of electrodes biased with opposite RF and DC voltages. The applied RF and DC voltages determine which mass will pass through the quadrupole filter. The detector then reads the current of the filtered charged particles with a specific mass, the intensity of which implies the amount of the species with that specific mass in the main chamber. The operation parameters of the QMS in this work are as follows: 70 eV ionization energy, 100 µA emission current, and 1900 V detector multiplying voltage.

Since QMSs only offer the intensities of the detected signals in arbitrary units, plasma diagnostics with QMSs require thorough modeling to obtain quantitative densities of gas-phase species. Nevertheless, they are still powerful in process monitoring since qualitative changes in signals are sufficient to monitor changes in processing plasma with the variation of external parameters such as power or pressure. Another strength of QMS diagnostics is that they are able to monitor plasma processing in real time. Plasma diagnostics inserting probes into the plasma, such as the cutoff and emissive probes, have a limitation for the in situ monitoring of material processing due to the shadowing of the plasma on materials induced by the probe insertion. In situ monitoring plays a particularly essential role in processes where the plasma dynamically changes with time such as in ALE.

## 3. Results and Discussion

Figure 5 plots the diagnostic results of the cutoff probe and emissive probe, showing changes in the electron density and plasma potential of Ar and Ar/C_4_F_8_ plasma with increasing RF power. Remembering that the ALE modification step changes from FC film deposition to continuous SiO_2_ etching as the RF power increases from 100 W to 300 W in the modification step (see Figure 2), the diagnostic results here show that the electron density increases from $1.6\times 10^{9}\;{\mathrm{cm}}^{-3}$ to $3.71\times 10^{10}\;{\mathrm{cm}}^{-3}$, approximately 20 times, while the plasma potential decreases from 30 V to 10 V, by two-thirds, which implies that for the ion energy flux, the particle’s number of ions significantly increases with slightly decreased energies. This trend of the changes in electron density and plasma potential is also found in Ar plasma in the removal step of the ALE process. The electron density increases from $1.48\times 10^{10}\;{\mathrm{cm}}^{-3}$ to $1.492\times 10^{11}\;{\mathrm{cm}}^{-3}$, approximately 10 times, while the plasma potential decreases from 15 V to 11 V, by one-third. According to the SiO_2_ etch model introduced above, an increase in ion energy flux leads to a higher decrease rate of material thickness [36]. It is thus possible that the different ALE results at the 100 W and 300 W levels of RF power stem from an increase in the ion energy flux in both Ar and Ar/C_4_F_8_ plasma, which is found via electron density and plasma potential, but only if the FC radical densities remain unchanged with the RF power increase. Therefore, FC radical diagnostic results should follow to evaluate the change in FC film deposition rates for a more rigorous model interpretation.

Figure 6 plots the results of FC radical density measurements using a QMS during one cycle of ALE at different RF powers of 100 W and 300 W. It is shown that during C_4_F_8_ injection, all of the polymeric radical species’ densities, except F, decrease with the increase in RF power, likely indicating that the FC film deposition rate is lower at 300 W than 100 W. Here, the increase in ion energy flux with increasing RF power elucidates the processing regime transition from FC film deposition to continuous etching during the surface modification step.

After the C_4_F_8_ cutoff, the FC radical densities instantaneously decrease regardless of RF power, as shown in Figure 6a–d. However, it is noticeable that the FC radical densities at 300 W do not fully return to the level before C_4_F_8_ injection but stop at an intermediate level and then slowly decrease. This may be attributed to the FC radicals absorbed into the chamber wall in the modification step being physically sputtered by ions bombarding the wall. Note that even at 300 W RF power, it is possible for FC films to form on the wall since there is no chemical etching between SiO_2_ and FC films, leading to a lower thickness decrease rate in the SiO_2_ etch model, unlike in the SiO_2_ surface where continuous etching occurs instead of FC film deposition in the modification step. Although the FC films on the wall that act as an FC radical source in the removal step also exist under the RF 100 W condition, the ion energy flux at 100 W may not be sufficient to induce an observable generation of FC radicals from the wall compared to that of the gas phase in the plasma (see Figure 5), leading to the full decrease in the FC radical densities after the C_4_F_8_ cutoff (note that the ion flux at 300 W is almost 10 times higher than that at 100 W).

We stress that the FC radical supply from the wall is considered to result in a significant difference in the thickness trends between 100 W and 300 W in the removal step shown in Figure 2; the thickness eventually saturates at 100 W, while it continuously decreases at 300 W. An increase in ion energy flux with increasing RF power leads the ion sputtering of FC films on the wall to no longer be negligible, driving it to act as an undesirable etchant source. Meanwhile, the effects of ion sputtering of the FC film deposited on the wall at 100 W give rise to infinitesimal drifts in the etched amount per cycle (EPC) as the ALE process proceeds (see Figure 2a). The increase in the EPC cycle by cycle may be led by an increase in the undesirable FC radical flux from the wall that accumulates cycle by cycle. This result implies that chamber wall conditioning needs to be carefully considered at certain processing conditions; more rigorous investigations into this issue will be conducted in future work.

Synthesizing these multiple plasma diagnostic results, we summarize that the process window shift with increasing RF power is caused by an increase in the electron density, a decrease in plasma potential, and a decrease in polymeric radical densities during the ALE process. The consistency between the changes in the process trend and the plasma parameters is in good agreement with the previously reported SiO_2_ etch model [36], and additional analysis considering the FC radical induced by ion sputtering of the deposited FC films on the wall well explains the continuous etching after the C_4_F_8_ cutoff at 300 W. Accordingly, the multiple plasma diagnostic methods in the present work are expected to be beneficial to establishing finely tuned ALE windows in the future.

## 4. Conclusions

As plasma processing has become widely employed in material processing, plasma diagnostic techniques play a bigger role in understanding and manipulating processing plasma for better outcomes. In the present work, the process window shift, where an increase in RF power pushes the processing condition out of the window, was investigated via multiple plasma diagnostic methods. Based on the previously reported SiO_2_ etch model, target species for the diagnostics of electron density, plasma potential, and FC radical densities were chosen. The obtained diagnostic results were able to sufficiently explain the process window shift, and in addition, were in good agreement with the etch model prediction.

It is worth mentioning that the utilization of multiple diagnostic tools to monitor the same plasma makes it easier to interpret the results of plasma processes, as shown in this work. Ultimately, for some complex plasma processes, such as ALE where plasma dynamically changes during the process, in situ plasma diagnostic methods are expected to offer more informative diagnostic results, allowing more precise and appropriate process controls.

## Figures and Tables

**Figure 1 sensors-22-06029-f001:**
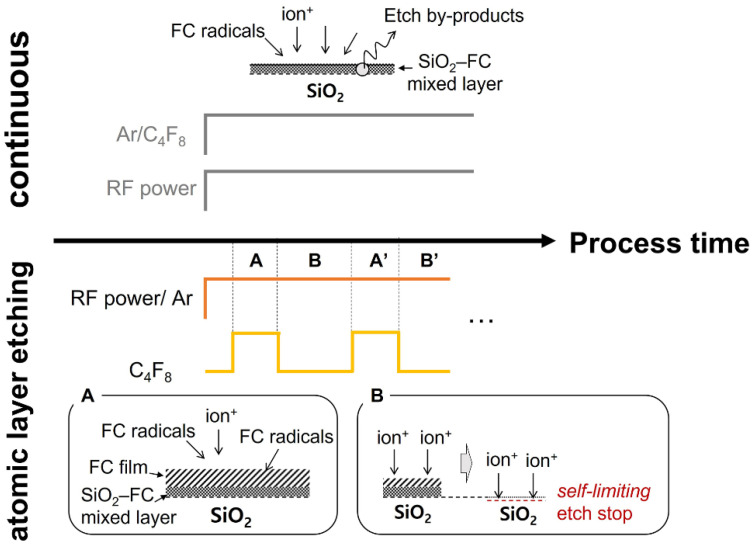
Comparison between continuous etching and ALE.

**Figure 2 sensors-22-06029-f002:**
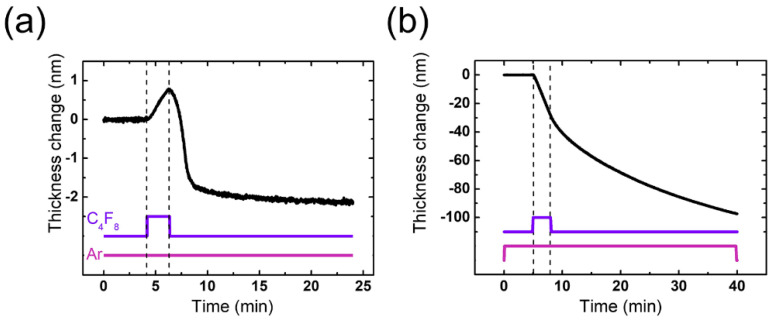
Thickness change in SiO_2_ during one cycle of ALE at (**a**) 100 W and (**b**) 300 W.

**Figure 3 sensors-22-06029-f003:**
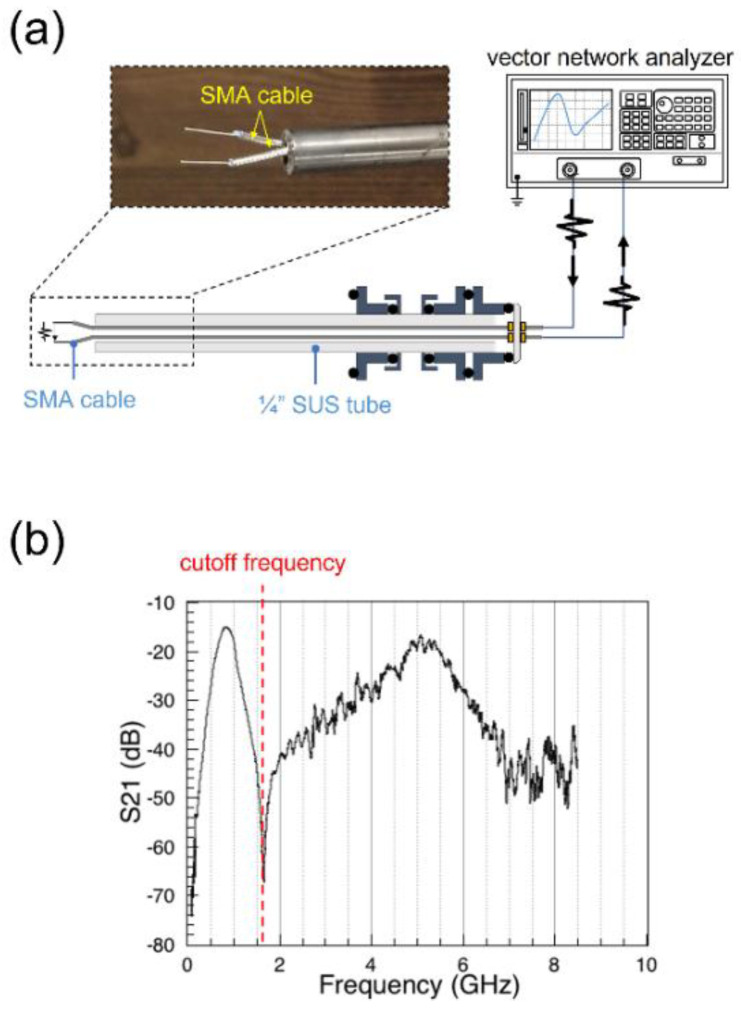
(**a**) Schematic of the cutoff probe measurement system and (**b**) characteristic S21 spectrum of the cutoff probe.

**Figure 4 sensors-22-06029-f004:**
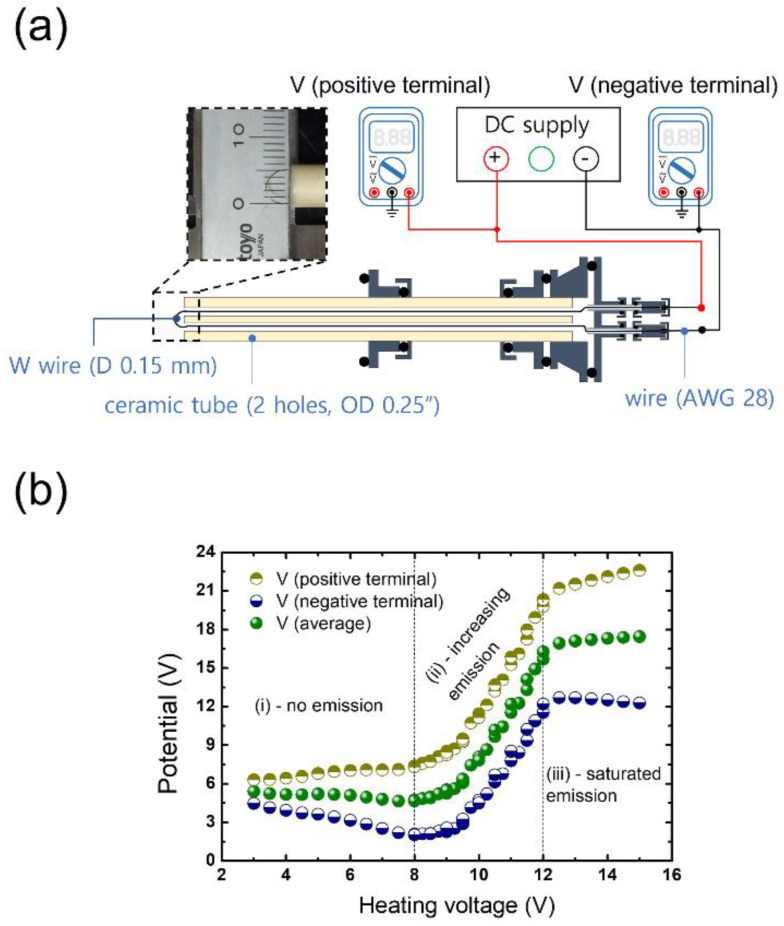
(**a**) Schematic of the emissive probe measurement system and (**b**) characteristic potential vs. heating voltage plot of the emissive probe.

**Figure 5 sensors-22-06029-f005:**
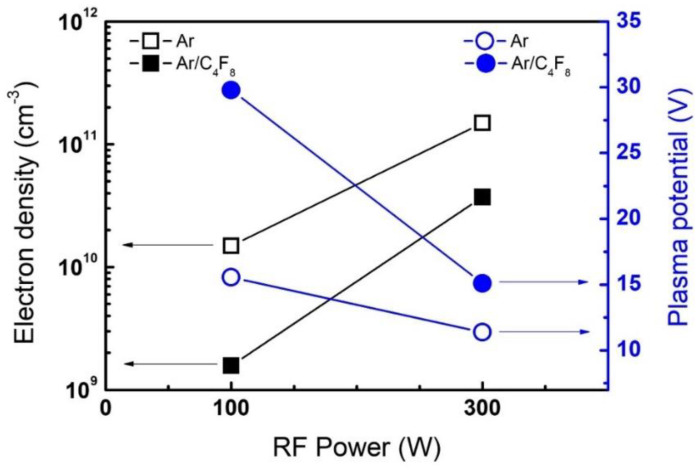
Changes in electron density and plasma potential with an increase in RF power from 100 W to 300 W in Ar and Ar/C_4_F_8_ plasma.

**Figure 6 sensors-22-06029-f006:**
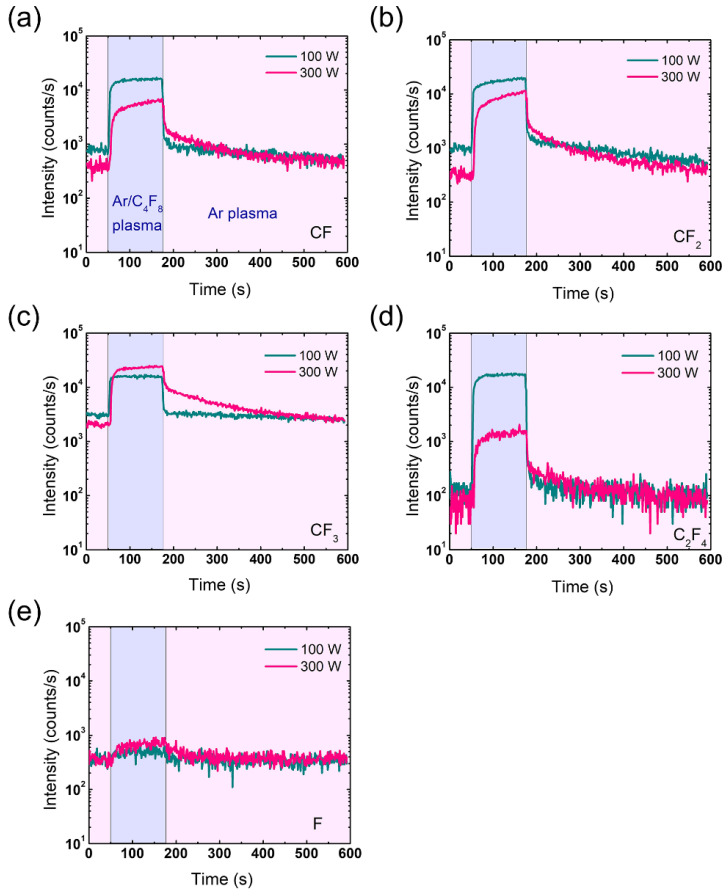
FC radical densities measured using a QMS during one cycle of ALE at 100 W and 300 W of (**a**) CF, (**b**) CF_2_, (**c**) CF_3_, (**d**) C_2_F_4_, and (**e**) F.

## Data Availability

The data presented in this study are available on request from the corresponding author.

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
