# Peer review of "Low-Temperature Plasma Diagnostics to Investigate the Process Window Shift in Plasma Etching of SiO2"

_sensors, 2022, doi:10.3390/s22166029_

Round 1

Reviewer 1 Report

I have read the article titled 'Low-temperature plasma diagnostics to investigate the processing window shift in plasma SiO2 etching' carefully but I regret that I have to recommend a rejection for the following reasons:

1. I don't see where the significance of the article really is and what novel contributions to the field it makes.

2. there are only a few measurements (too few for an experimental paper) and the ones that are presented are not really meaningful to me. E.g. Fig. 2 shows a comparison between different etching methods under different parameters - this is like comparing apples to oranges. It is completely unclear what the influence of the parameter changes is. If the authors want to compare different procedures, they should change only one parameter at the time and explain clearly why this was done and what the meaning is.

3. The measurements were only done for two input powers, which does not yield any information at all, no trends, no qualitative or quantitative behaviour. Why this was done I don't know because obtaining more measurement points would not be that much of an effort.

4. The same holds for the plasma potential with the emissive probe. In addition the presentation of the data is very strange. What one usually does is plotting the emissive probe bias vs. the current for a strongle emissive probe (as it is even done in the references given by the authors). Then the floating point is determined as the plasma potential since there is plenty of literature that corroborates why this is possible. However, the authors plotted the heating voltage vs. the potential, whatever the reason for this is. The main point of using an emissive probe is that is easy to determine the plasma potential from the floating point but in Fig. 4 b) there is no floating point (i.e. the point where the probe current is zero).

5. The axis titles of Fig. 7 are so small that they are barely readable but as far as I can see the change in CF3 is comparable to all the other radicals (except F). To state in the text that it barely changes is simply wrong.

6. The rest of the text is much basic content about why and how to use MS, probes, etc... it is not very specific and after reading it I am still not sure what the aim of the paper really is.

Thus, I have regretfully recommend a rejection of the paper.

Author Response

Point 1: I don't see where the significance of the article really is and what novel contributions to the field it makes.

Response 1: We appreciate your rigorous and critical review. The meaning of the article is believed to be in the successful interpretation of the processing window shift via plasma diagnostics with collaboration from the previously-reported SiO2 analytic etch model. Understanding the underlying mechanism in processing window shifts seems especially important to plasma processes such as atomic layer etching where the precise control of processing conditions is critical. The well implement of the model to plasma processing interpretations requires appropriate plasma parameters as the original paper where the SiO2 etch model has been first suggested demonstrates. Thus, it should be pointed that the submitted article shows an extended employment of the SiO2 etch model to more intricate plasma processing where different input powers and gas flows can make notable differences in the results, which is allowed by the synthetical implement of plasma diagnostic methods.

Point 2: There are only a few measurements (too few for an experimental paper) and the ones that are presented are not really meaningful to me. E.g. Fig. 2 shows a comparison between different etching methods under different parameters - this is like comparing apples to oranges. It is completely unclear what the influence of the parameter changes is. If the authors want to compare different procedures, they should change only one parameter at the time and explain clearly why this was done and what the meaning is.

Response 2: We appreciate your comment. As the comment points out, what is intended in Fig. 2 is to show the process window shift from FC film deposition to continuous etching, which is solely caused by an increase in RF power. It is worth noting that in SiO2 ALE with fluorocarbon plasma, achieving FC film deposition conditions is essential for the surface modification. Otherwise, ALE simply becomes conventional etching. Therefore, in this case, we believe that comparing simply between the two typical trends, deposition and etching, would be more effective to introduce the process window shift. Since the purpose of the article is to investigate the causes of processing window shifts via plasma diagnostics, we aim to exhibit only the typical trends of atomic layer etching and continuous etching in Fig. 2 prior to the explanations of the used plasma diagnostic methods as an introduction of the following main discussion.

Point 3: The measurements were only done for two input powers, which does not yield any information at all, no trends, no qualitative or quantitative behavior. Why this was done I don't know because obtaining more measurement points would not be that much of an effort.

Response 3: We appreciate your comment. We completely agree with the comment that more data points would provide more informative diagnostic results. The reason why the measurements were done only for two input powers is because the purpose of the diagnostics in this work is to investigate plasmas in two distinct conditions like apples and oranges, not green apples turning into red. We thus consider it to be sufficient to diagnose plasma parameters specifically under atomic layer etching and continuous etching conditions represented in Fig. 2.

Point 4: The same holds for the plasma potential with the emissive probe. In addition the presentation of the data is very strange. What one usually does is plotting the emissive probe bias vs. the current for a strongle emissive probe (as it is even done in the references given by the authors). Then the floating point is determined as the plasma potential since there is plenty of literature that corroborates why this is possible. However, the authors plotted the heating voltage vs. the potential, whatever the reason for this is. The main point of using an emissive probe is that is easy to determine the plasma potential from the floating point but in Fig. 4 b) there is no floating point (i.e. the point where the probe current is zero).

Response 4: Thank you for your critical comment. In this work, we employed some special emissive probe diagnostic method that has been developed to measure fast time evolutions of the plasma potential (DOI: 10.1143/JJAP.22.148); when a non-emissive probe, a electrically-floating metal rod with an unheated filament, is in contact with plasma, it has a potential lower than the plasma potential by a factor of the floating sheath potential. As described in the manuscript, heating the probe filament causes thermionic electron emissions that reduce the floating sheath potential, allowing the emissive probe to be almost equal to the plasma potential. Following the above mechanism, we could measure the plasma potential by simply measuring the potential of the emissive probe with respect to the ground.

Point 5: The axis titles of Fig. 7 are so small that they are barely readable but as far as I can see the change in CF3 is comparable to all the other radicals (except F). To state in the text that it barely changes is simply wrong..

Response 5: We appreciate your meticulous review. Following the comment, Fig. 7 and the wrong description the comment points out have been revised (this revision is represented in blue in the revised manuscript).

Point 6: The rest of the text is much basic content about why and how to use MS, probes, etc... it is not very specific and after reading it I am still not sure what the aim of the paper really is.

Response 6: We appreciate your critical comment. It is great regret that the article hasn’t sufficiently provided convincing explanation. As replied in Response 1, the main argument of the article is to identify the underlying mechanism in processing window shifts via plasma diagnostics with collaboration with the SiO2 etch model, which is especially meaningful for atomic layer etching. The authors will gladly revise the manuscript if there still were a chance to receive any reviewer comments, no matter how major they are, and will sincerely appreciate if the article would be reconsidered for publication.

Reviewer 2 Report

The authors investigated the causes of processing window shifts with various plasma diagnostic tools. Cutoff and emissive probes, quadrupole mass spectrometry, and ellipsometry are used to explain the process window shift. In situ plasma diagnostic methods in current work can offer informative diagnostic results. It is very interesting study and worthy to publish for Sensors journal. However, it needs minor correction as below.

1. The size of character in figures is too small to recognize easily.

2. In Figure 6 (in results and discussion part), the plasma potential and the electron density are shown. However more detail process will be informative to reader. Please describe experimental results and the calculations of those values.

3. The radical intensities for 100 W are higher than for 300 W with C4F8 injection in Figure 7. Is this phenomenon is also related to processing window shift? Is it possible to explain window shift in comprehensive manner with the radical intensities, the plasma potential, and the electron density?

Author Response

Point 1: The size of character in figures is too small to recognize easily.

Response 1: We appreciate your meticulous review. Following the comment, all figures have been reviewed and revised for better visibility.

Point 2: In Figure 6 (in results and discussion part), the plasma potential and the electron density are shown. However more detail process will be informative to reader. Please describe experimental results and the calculations of those values.

Response 2: We appreciate your kind comment. The description has been revised to include specific values of the measurement results (this revision is represented in blue in the revised manuscript).

Point 3: The radical intensities for 100 W are higher than for 300 W with C4F8 injection in Figure 7. Is this phenomenon is also related to processing window shift? Is it possible to explain window shift in comprehensive manner with the radical intensities, the plasma potential, and the electron density?

Response 3: With increasing RF power from 100 W to 300 W, the radical intensities decrease as mentioned in the comment, leading to lower FC film deposition rates. Whereas, the ion energy flux, which is a multiplication between the ion density and the plasma potential, increases with the same RF power change, creating higher etch rates of FC films and SiO2. Thus, the window shift from FC film deposition to continuous etching at the modification step can be interpreted with these diagnostic parameters in a comprehensive manner.

Reviewer 3 Report

A more common plasma diagnostic used in plasma processing tool to detect process drift is OES.  The author may want to explain why OES is not adopted while using other dignostics that are more complex and cost more.   Another issue that is also very important and need to be addressed is the contamination caused by metallic probe tips.   

Author Response

Point 1: A more common plasma diagnostic used in plasma processing tool to detect process drift is OES. The author may want to explain why OES is not adopted while using other dignostics that are more complex and cost more. Another issue that is also very important and need to be addressed is the contamination caused by metallic probe tips.

Response 1: We appreciate your kind comment. As for OES, it is one of the most widely used plasma diagnostic tools. However, it may not be appropriate to measure plasma parameters under various conditions and employ the results to analytic model-based interpretations as done in this work since OES measurements of, for example, radical intensities depend not only the absolute density of the radicals but also the electron density that allows the radicals to emit light with radicals’ characteristic wavelength. Thus, we employed other diagnostic tools that directly show somewhat quantitative results of measured parameters. In addition, the issue metallic probe tip contamination has been considered in this work and this provoked the use of the cutoff probe and the emissive probe since both probes have been reported to have strong resistance on polymer contamination in their measurements. This is briefly introduced in the manuscript in Section 2.2.1. and 2.2.2. (the related descriptions are represented in blue in the revised manuscript).

Reviewer 4 Report

In my opinion, the paper is not suitable for “Sensors” and should be transferred to another journal.  Major revision required.

Title: use: “plasma etching of SiO2“

Title, and throughout paper: “process window” ?

Line 42: replace “bigger” by “large”

Figures, general remark: Figures and, in particular, labels are too small. Enlarge figures too full text width.

Figs. 3(a) and 4(a): labels are washy (not sharp).

Fig. 5: (a) delete, is common knowledge. (b): what is shown here, a residual gas spectrum? Too much water, no Ar and no FC noticeable.    

SiO2 etch model (e.g., line 42, and elsewhere). Does it refer to Fig. 1? Can you use/show (reaction) formulae?

Line 101: mixture of Ar and C4F8, what is the purity of these gases? Which isomer of C4F8 was used?  

Line 127: mTorr is outdated. Use “Pa” instead.

Line 153: increasing and decreasing rates or something “growth” and “loss” rate? I would prefer if you show some kind of rate equation.

Line 210: what is a S21 spectrum? How are “cut-off” frequency and “plasma” frequency connected?

Figure 7 shows measured radical densities. Where are these radicals produced? It is well known that fragmentation of parent molecules not only occurs in the plasma but also during ionisation in the mass spectrometer’s ion source; see NIST Chemistry Webbook, https://webbook.nist.gov/cgi/cbook.cgi?ID=C115253&Units=SI&Mask=200

For example, the parent molecule ion C4F8+ does not even exist.  

Author Response

Point 0: In my opinion, the paper is not suitable for “Sensors” and should be transferred to another journal. Major revision required.

Response 0: Thank you for your comment. The aim of the article is to figure out the cause of process window shifts that occur during an external parameter variation via the comprehensive employment of plasma diagnostic methods. Although trends of plasma process can be predicted, at least some part, with the analytic model employed in this work, more rigorous interpretation requires reliable plasma diagnostic measurements. Since what is done in this work is sensing the changes in plasma according to external parameters and translating them to physical quantities of plasma to interpret process window shifts, we believe that the article is well suitable for the special issue of Sensors entitled “Plasma Diagnostics”.

Point 1: Title: use: “plasma etching of SiO2”. Title, and throughout paper: “process window”?

Response 1: Thank you for your comment. The expressions recommended in the comment have been applied through the entire manuscript (the revisions are represented in blue in the revised manuscript).

Point 2: Line 42: replace “bigger” by “large

Response 2: Revised accordingly. (the revision are represented in blue in the revised manuscript).

Point 3: Figures, general remark: Figures and, in particular, labels are too small. Enlarge figures to full text width. Figs. 3(a) and 4(a): labels are washy (not sharp).

Response 3: Thank you for your meticulous review. All figures have been revised for better visibility.

Point 4: Fig. 5: (a) delete, is common knowledge. (b): what is shown here, a residual gas spectrum? Too much water, no Ar and no FC noticeable.

Response 4: Thank you for your kind comment. Despite its commonness, it is considered that keeping Fig. 5(a) may make the manuscript more organized on the parallelism with Fig. 3 and 4. If the reviewer doesn’t mind, the authors prefer for Fig. 5(a) to remain. Fig. (b) is introduced just as an example of a measured mass spectrum by QMS obtained under low vacuum conditions. The measurement shown in Fig. 5(b) has nothing to do with the following radical measurements.

Point 5: SiO2 etch model (e.g., line 42, and elsewhere). Does it refer to Fig. 1? Can you use/show (reaction) formulae?

Response 5: Thank you for your comment. The SiO2 etch model refers to SiO2 etching with FC plasmas so that the model covers the continuous etching and the step A in atomic layer etching. Following the comment, the model formulae has been inserted in the introduction section (the revisions are represented in blue in the revised manuscript).

Point 6: Line 101: mixture of Ar and C4F8, what is the purity of these gases? Which isomer of C4F8 was used?

Response 6: Thank you for your comment. The purity of both Ar and C4F8 is five nines (99.999%). The name of the C4F8 isomer used in this work is octafluorocyclobutane.

Point 7: mTorr is outdated. Use “Pa” instead.

Response 7: Thank you for your kind comment. The unit “mTorr” has been revised to “Pa”.

Point 8: Line 153: increasing and decreasing rates or something “growth” and “loss” rate? I would prefer if you show some kind of rate equation.

Response 8: Thank you for your kind comment. As mentioned in the comment, “increasing” and “decreasing” means “growth” and “loss”, respectively. Following the comment, the rate equation, which is referred to “the SiO2 etch model” in the article, has been inserted in the introduction section (the revisions are represented in blue in the revised manuscript).

Point 9: Line 210: what is a S21 spectrum? How are “cut-off” frequency and “plasma” frequency connected?

Response 9: Thank you for your comment. A S21 spectrum can also be termed as a transmission spectrum. S21 refers to the intensity ratio between incident waves from a wave port 1 and detected waves at a port 2. If microwaves with frequencies lower than the plasma frequency is incident into the plasma, they will be cut off soon since plasma acts like metals for microwaves with frequencies lower than the plasma frequency that refers to the dynamicity of plasma electrons. The explanation on the cutoff phenomenon is briefly described in the section 2.2.1. (the related description is represented in red in the revised manuscript).

Point 10: Figure 7 shows measured radical densities. Where are these radicals produced? It is well known that fragmentation of parent molecules not only occurs in the plasma but also during ionisation in the mass spectrometer’s ion source; see NIST Chemistry Webbook, https://webbook.nist.gov/cgi/cbook.cgi?ID=C115253&Units=SI&Mask=200. For example, the parent molecule ion C4F8+ does not even exist.

Response 10: Thank you for your critical comment. We have seriously considered the issue addressed in the comment, yet it remains a problem to be solved. Since the ionizing electron energy used in QMS of this work is 70 eV that is strong enough to create dissociative ionizations of C4F8, it may lead to unfavorable noise signals as the comment points out, making absolute density measurements unreliable. However, we believe that when comparing only relative intensity differences obtained under different conditions, such noises from dissociative ionizations are expected to comparably affect the measurements under different conditions, possibly remaining relative comparisons acceptable.

Round 2

Reviewer 1 Report

Dear editor,

I am afraid the responses of the authors are still not satisfying. The authors somewhat addressed my previous point 1 but did not put much of this additional information into the article. The answer to my point 2 was also not satisfying because the film thickness curve in Fig. 2 does not only change due to an increase of rf power as the authors claim but also due to the repeated introduction of C4F8 into the process chamber at regular time intervalls. Hence, the authors change at least two process parameters at the same time while only measuring once.

The same holds for my point 3.

Point 4 has not really been addressed at all. I have to add that I did my fair share of emissive probe measurements myself and still this Fig. 4 b is not a usefull representation of the data. Plotting the heating voltage against the probe potential is meaningless. One would have to show the probe potential vs. the probe current in order to get useful results. Also the new paper that the authors provide in order to corroborate their choice of probe data measurements does not really anything to help their point. Nowhere in this new reference is a plot or a method leading to a plos such as Fig. 4 b described.

Point 5 has been addressed satisfactorily but this was only a very minor point, which has no bearing on the scientific merrit of the paper.

Point 6 was made to let the authors know that there is just a lot of text that explains basic concepts of the measurements applied, which has no direct bearing on the measurement outcome. In principle, it is positive if a short paragraph or two is/are added to explain the applied measurement principles but in this case it was also too much text book knowledge for a science paper in the first version. The authors 'fixed' this by adding even more basics about the plasma frequency and the penetration of EM waves into the plasma, etc...

Hence, regrettfully I have still to recomend a rejection of the paper.

Author Response

Dear reviewer 1,

We regret that the previous responses were not satisfying. Here, we shortly make up the last weak responses.

At first, we address again the point 1 in the last reviewer comments about the significance and novelty of the article by adding a new line to the end of the abstract as follows:

“By extending the SiO2 etch model with numerous diagnostic methods, more intricate plasma processing conditions can be characterized, which will be beneficial in applications and industry where different input powers and gas flows can make notable differences in the results.”

After seeing the comment on Fig. 2 above, we finally understand what was the point of the point 2 in the last reviewer comment. We intended in Fig. 2 to show the comparison between the results in ALE windows and not in ALE windows (plotted in Fig. 2(a) and 2(b), respectively). However, the mistake we missed was that three cycles of ALE were plotted in Fig. 2(a) while only one cycle of non-ALE was shown in Fig. 2(b), which would eventually make readers confused, as the comment points out. The purposed comparison in Fig. 2 is between the etch trend of the first cycle of ALE in Fig. 2(a), which occurs with one C4F8 injection and cutoff, and the continuous etch trend in Fig. 2(b) despite of the same one C4F8 injection and cutoff. The second and third cycles of ALE plotted in Fig. 2(a) are intended to highlight that subsequent ALE processes proceed well in the ALE process window. In short, what Fig. 2 is supposed to show is that the different etch trends during both C4F8 injection and cutoff periods from between ALE windows and non-ALE windows. To clarify the purpose, we revised Fig. 2(a) to show the result of only one ALE cycle.

This also explains why we conducted plasma diagnostics under only two RF power conditions, 100 W and 300 W. Since the process window shift occurs only with RF power variation and the results at 100 W and 300 W already exhibit two totally different etch trends, we believe that it is sufficient to conduct plasma diagnostics only under their process conditions.

As for the emissive probe diagnostic method used in this work, we introduce another reference (DOI: 10.1063/1.1137486) that directly includes the plot of the measured floating potential vs the filament voltage, as Fig. 4(b) in the manuscript describes, and the related figure is below.

Lastly, we want to stress that the main object of the article is to identify the underlying mechanism in processing window shifts via plasma diagnostics with the use of the SiO2 etch model, which is especially meaningful for atomic layer etching. This purpose is now better highlighted for readers in the revised manuscript following all the suggestions by the reviewer. In terms of the “basic” content, it was the authors’ intention to provide a somewhat wide view in this paper to give readers of this special issue some necessary background.

Reviewer 4 Report

Figs. 5(a) and (b) make no sense at all and should be removed. 

Author Response

Point 1: Figs. 5(a) and (b) make no sense at all and should be removed.

Response 1: Following the comment, Fig. 5 has been removed (all the numbers of the following figures and captions have been adjusted in the revised manuscript).

Round 3

Reviewer 1 Report

The authors made some improvements regarding the presentation of the data but I still have to say that I do not see the additional scientific value of the paper and the emissive probe chart is still not clear to me. Granted, that the authors found a paper where the probe voltage that is proportional to the emissive probe's temperature to some extend is plotted against the floating potential. However, as I stated before that is quite unusual because the more relevant parameter would be the probe temperature, which would have to be calculated from the probe heating voltage, current and the Richardson-Duschman equation. Even if I assume that the saturation of the voltage curve is indeed converging against the plasma potential, I don't see the obtained value of 18 V (as shown in Fig. 4) anywhere in the results presented in Fig. 5. Thus, I am not convinced that the authors have applied their emissive probe measurements correctly, which possibly renders a very important section of the paper invalid.

Author Response

The response letter contains several figures and equations, which may not be allowed  in the box. Thus,
